# New technology for an ancient fish: A lamprey life cycle modeling tool with an R Shiny application

Dylan G. E. Gomes[1]*, Joseph R. Benjamin[2], Benjamin J. Clemens[3],
Ralph Lampman[4], Jason B. Dunham[5]

1 United States Geological Survey, Forest and Rangeland Ecosystem Science Center, Seattle, Washington, United States of America, 2 United States Geological Survey, Forest and Rangeland Ecosystem Science Center, Boise, Idaho, United States of America, 3 Oregon Department of Fish and Wildlife, Corvallis Research Lab, Corvallis, Oregon, United States of America, 4 Yakama Nation Fisheries Resource Management Program, Toppenish, Washington, United States of America, 5 United States Geological Survey, Forest and Rangeland Ecosystem Science Center, Corvallis, Oregon, United States of America

* dylan.ge.gomes@gmail.com

## Abstract

Lampreys (Petromyzontiformes) are an ancient group of fishes with complex life histories. We created a life cycle model that includes an R Shiny interactive web application interface to simulate abundance by life stage. This will allow scientists and managers to connect available demographic information in a framework that can be applied to questions regarding lamprey biology and conservation. We used Pacific lamprey (*Entosphenus tridentatus*) as a case study to highlight the utility of this model. We applied a global sensitivity analysis to explore the importance of individual life stage parameters to overall population size, and to better understand the implications of existing gaps in knowledge. We also provided example analyses of selected management scenarios (dam passage, fish translocations, and hatchery additions) influencing Pacific lamprey in fresh water. These applications illustrate how the model can be applied to inform conservation efforts. This tool will provide new capabilities for users to explore their own questions about lamprey biology and conservation. Simulations can hone hypotheses and predictions, which can then be empirically tested in the real world.

## Introduction

Lampreys are a widespread group of ecologically and culturally important fishes, with many species of conservation concern [1–5]. Lampreys represent an 'ancient' lineage, branching from the ancestors of modern jawed vertebrates between 450 and 550 million years ago [6–8], with body forms that have remained mostly unchanged for at least 360 million years [9,10]. Given their unique evolutionary history, potential losses of lampreys represent a threat to the overall loss of the evolutionary diversity

**Data availability statement:** This submission uses novel code, which is provided, per our requirements, in an external repository to be evaluated during the peer review process (https://doi.org/10.5066/F7CV4H1T). No data were collected for this study. Full code citation: Gomes DGE. Lamprey life cycle model: U.S. Geological Survey Software Release. 2024. doi: https://doi.org/10.5066/F7CV4H1T.

**Funding:** This work was funded by the Bonneville Power Administration (BPA; https://www.bpa.gov/; Project Number 2017-005-00; BPA contract #93262), U.S. Department of Energy, as part of BPA's program to protect, mitigate, and enhance fish and wildlife affected by the development and operation of hydroelectric facilities on the Columbia River and its tributaries. The views in this report are the author's and do not necessarily represent the views of BPA. The BPA did not play any role in the study design, data collection and analysis, decision to publish, or preparation of the manuscript.

**Competing interests:** The authors have declared that no competing interests exist.

of fishes [11–13]. However, the formal status of many species of lamprey suffers from limited data availability [5].

One major difficulty in studying lampreys is the complexity and cryptic nature of their life cycles (Fig 1). For example, lampreys are photophobic and their larval life stage reside within the substrate for years [14]. Additionally, lampreys undergo substantial transformations between larval and adult stages, which can include anadromous or adfluvial migration behaviors in large bodies of water, such as oceans and lakes [3,15]. These life stages are difficult to study due to the sparseness of animals in such pelagic environments and the consequent high cost of data collection. Furthermore, within a single lamprey species, considerable variation in environmental conditions is experienced [3,15], necessitating a flexible tool to explore the understudied aspects of lamprey biology. Few reliable methods exist for studying lampreys throughout their life cycle [16–18].

Life cycle models can be valuable tools for exploring dynamics across the entire life cycle of an organism. Parameters such as fecundity and survival at each life stage are typically set based on central tendencies (e.g., mean, median), and can include the associated uncertainty values (e.g., standard error, standard deviation, 95% confidence intervals, range) with such estimates. However, gaps in knowledge need not delay the development of such holistic modeling frameworks. Instead, users can leverage computer simulations to assess likely values of unknown parameters to determine how influential these parameters are in influencing model outputs, which can in turn guide future research or monitoring needs. Such tools can also be applied effectively to identify critical uncertainties and consequences of decision alternatives, even if parameter values are not entirely certain [19].

Here, we present a Lamprey Life Cycle Model (LLCM) [20], in a simulation-based framework, to explore the sensitivities and knowledge gaps in the lamprey life cycle and explore examples of management scenarios. For example, the LLCM that we present allows for the exploration of lamprey population sensitivity to life cycle parameter values and uncertainties so that users can evaluate which parameters are the most important to refine via future empirical data collection efforts. The LLCM also allows users to simulate and thereby assess the efficacy of various management actions and run a virtually infinite number of scenarios. Users can add any number of barriers (with the ability to adjust each barrier passage probabilities), add fish translocation efforts, add hatchery releases through larval and juvenile stages, and simulate mortality and carrying capacities during extreme climate events, such as low summer flows and drought and/or winter scouring, flooding, or freezing (Fig 2). We built this stage-based LLCM [20] in R [21] with an interactive user web interface in the R package Shiny [22], along with a package of model code for higher throughput analyses. Our model and tool [23] has the potential for continued modification to fit the various uses of the lamprey management community. As a specific case-study to demonstrate potential uses and benefits of such a modeling framework, we focus on using published information and best professional knowledge to inform parameter estimates for the Pacific lamprey, *Entosphenus tridentatus*. For other applications, lamprey managers and researchers

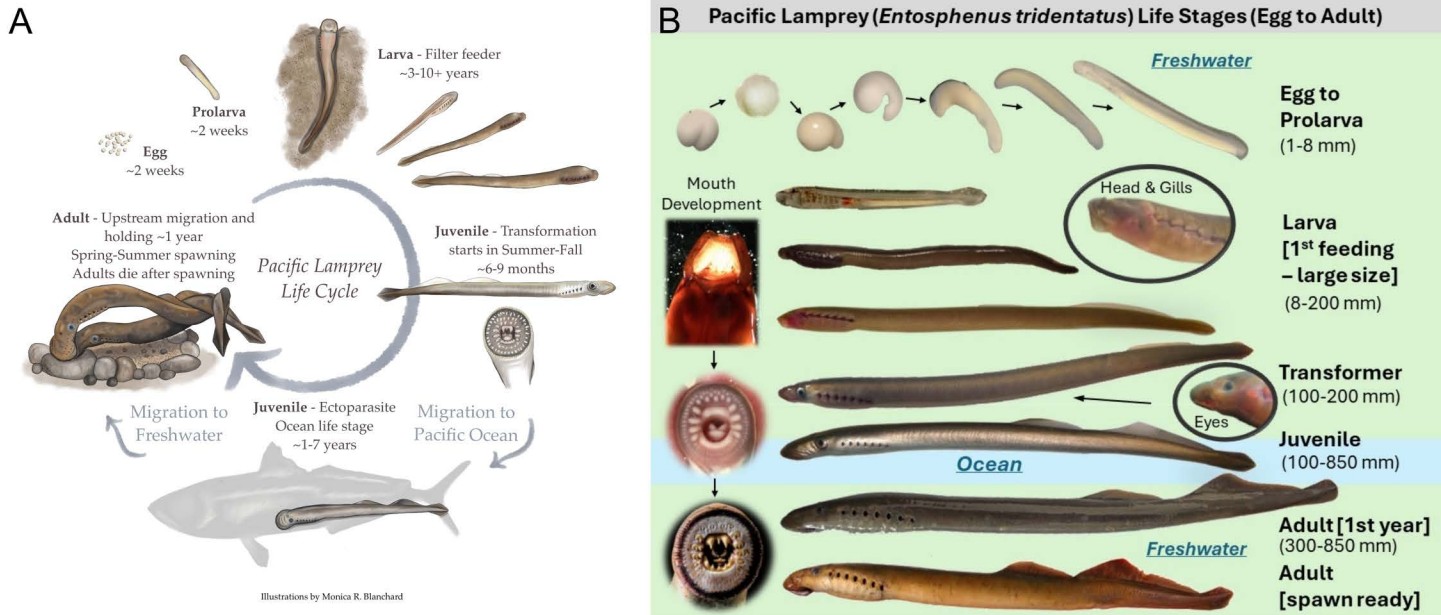

**Fig 1. (A) Artistic diagram of the Pacific lamprey anadromous life cycle.** Artwork by Monica R. Blanchard. (B) Images depicting Pacific lamprey life stages with details on notable traits at various stages of development. Images by Ralph Lampman.

can adjust parameter values based on empirical data and expert opinion to represent a life cycle model for any species of lamprey.

## Materials and methods

The model framework is built on the basic life cycle of parasitic, anadromous lampreys and simulates abundance across six serial life stages: spawning adults, eggs, larvae, transformers, juveniles, and returning adults, which spawn in freshwater ([24]; Fig 1). Fish move sequentially through each life stage and age class (Fig 1) and are subjected to survival processes including density-dependent survival (Fig 2). For simplicity, fish growth, among other physiological processes, is not currently considered. To add demographic stochasticity, and to explore parameter uncertainty, processes include draws from probability distributions (see below). Process-based models, such as this one, often consist of repeated simulation runs with stochastic sampling from probability distributions or multiple variants of a base model (i.e., Monte Carlo models), which allows parameter uncertainty to compound in model outputs and to propagate over time and/or space for the population or ecosystem of interest.

We used Pacific lamprey as a model species for the design and implementation of this life cycle model. The Pacific lamprey is an anadromous species of fish inhabiting a widespread area across the Northern Pacific Ocean and associated tributaries [3,15,25]. They are semelparous (die after spawning), construct redds (nests) in stream substrate, and spawn in freshwater streams [26,27]. Once eggs hatch and absorb its yolk sac (as prolarva), larval offspring burrow in sediment and filter feed for 2–10 years, depending on environmental variables such as stream temperature, gradient, and lamprey density [14,28–30]. After this time, eyeless larvae transform into eyed juveniles and migrate to sea, where they become ectoparasites upon other fishes and marine mammals for an additional 1–10 years [25,31,32]. Once juveniles begin their journey upriver they are considered adults [24], and generally spawn within 2 years of returning to fresh water [30,33–35].

Since we have chosen an anadromous species to represent the model, we refer to the "ocean" to indicate migratory destinations throughout the rest of this documentation (e.g., juveniles exist in the ocean). This designation could be

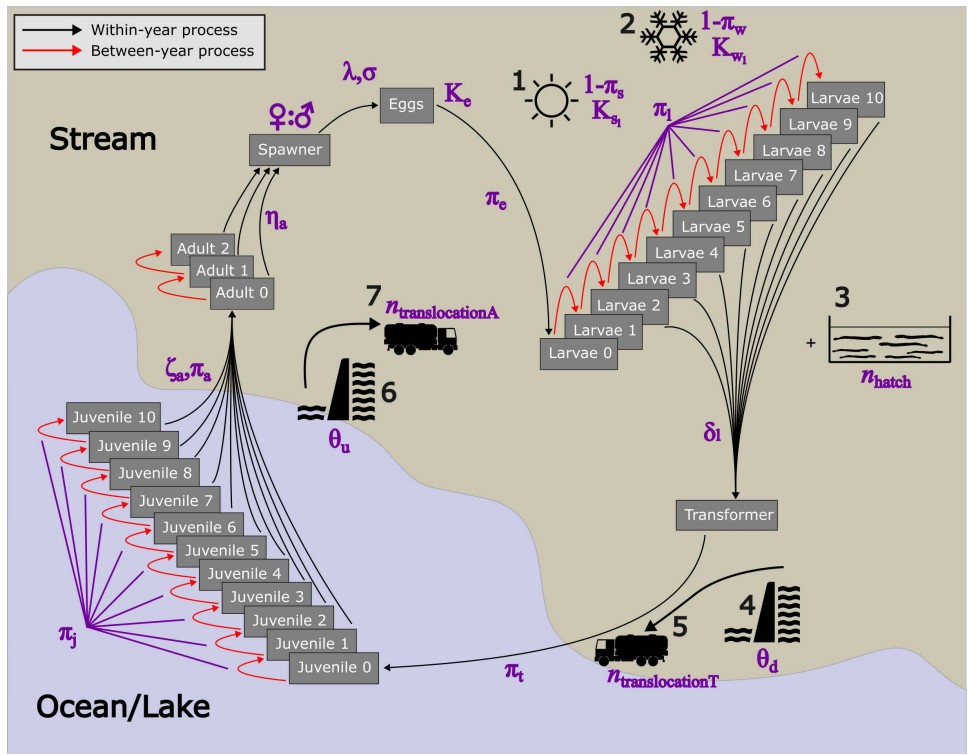

**Fig 2. Conceptual diagram of the Lamprey Life Cycle Model (LLCM).** Grey boxes indicate specific life-stage and age-class combinations. Black arrows indicate processes that occur within a given year (e.g., reproduction, transformation, etc.), while red arrows indicate processes that occur between years (e.g., aging). Purple symbols indicate life cycle parameters (see Table 1). Black icons indicate potential management scenarios to explore (1 = summer [drought] mortality and carrying capacity, 2 = winter [flood/scouring] mortality and carrying capacity, 3 = hatchery additions, 4 = downstream barriers, 5 = downstream [transformer] translocation, 6 = upstream barriers, 7 = upstream [adult] translocation).

adapted to migratory species that move to different destinations. For modeling other lamprey species of interest (e.g., sea lamprey, *Petromyzon marinus*), the "ocean" labels can simply represent bodies of fresh water, such as the Great Lakes. The user only needs to replace the relevant life cycle parameters in the model to represent different species of lamprey (e.g., sea lamprey [36]). Below, we describe each component in the lamprey life cycle model in sequence, starting with the transition from spawning adults to eggs, and following the life cycle through returns of adults to spawn.

**Life cycle model formulation**

**Spawners to eggs.** Survival is represented by the transition of individuals from one life stage to the next. The transition between the spawner and egg life stages includes fecundity in addition to egg survival. The number of eggs, $n_{egg}$, for each individual, $i$, is drawn from a negative binomial distribution with mean $\lambda_{\text{fecund}}$ and standard deviation $\sigma_{\text{fecund}}$:

$$n_{\text{egg}_i} \sim \text{negative binomial}\left(\lambda_{\text{fecund}}, k_{\text{fecund}}\right),$$

(1)

where $\lambda_{\text{fecund}}$ equals the mean fecundity and the negative binomial shape parameter, $k_{\text{fecund}}$, is related to the standard deviation via moment matching [37]:

$$k_{fecund} = \frac{\lambda^2_{fecund}}{\sigma^2_{fecund} - \lambda_{fecund}}.$$

(2)

Total eggs, $n_{egg}$, are summed across all spawners, where the number of spawners is the product of the total number of adults and the female sex ratio:

$$n_{egg} = \sum_{i=1}^{n_{spawners}} n_{egg_i}, \tag{3}$$

$$n_{spawners} = n_{adults} \cdot [\female : \male]. \tag{4}$$

**Eggs to larvae.** The Beverton-Holt function was used to account for density dependence in the freshwater survival of eggs [38]. The calculated survival estimates, $\pi_{egg}$, are then used as probabilities in a binomial distribution to incorporate demographic stochasticity to the simulated number of individuals at the subsequent age-0 larval life stage:

$$\pi_{egg} = \frac{1}{\left(\frac{1}{S_{egg}} + \frac{n_{egg}}{K_{egg}}\right)}, \tag{5}$$

$$n_{larvae_0} \sim \text{binomial}\left(n_{egg}, \pi_{egg}\right), \tag{6}$$

where $n_{larvae_0}$ is the number of age-0 larvae; $\pi_{egg}$ is the Beverton-Holt estimated, density-dependent, egg to larva survival probability; $S_{egg}$ is the baseline egg to larva survival, without density dependence (defined by the user); $n_{egg}$ is the number of eggs present, and $K_{egg}$ is the carrying capacity of eggs in the system or spawning capacity (in # eggs; defined by the user).

**Hatchery additions.** Hatchery releases occur during this step, before fish experience density dependence in the wild (see next step). The user can select any number of hatchery releases consisting of any number of age-1 to age-10 larvae and transformers. For simplicity, the model treats hatchery fish as externally-sourced (i.e., does not account for in-system hatchery production logistics such as removing fish for brood), and fish are added directly to the existing population. That is, the model assumes no differences between hatchery released fish and naturally occurring fish. Studies indicate that hatchery lamprey are largely comparable to wild counterparts in terms of behavior and physiology, yet subtle differences that render them unsuitable as representative replacements for run of the river lamprey may exist and this will need to be investigated further [39–41].

**Annual larval survival.** Larval survival occurs through density-independent (see below) and density-dependent processes. Density-dependent survival is calculated across all age-1+ larvae, regardless of size, through modifications of mortality ($M$, optional) and carrying capacity, $K$, during two sequential periods, summer and winter.

First, we sum larvae across all age-1+ larvae available at the beginning of the year (in the spring):

$$n_{spring.larvae} = \sum_{l=1}^{10} n_{larvae_l}. \tag{7}$$

We can specify any summer mortality, $M$, here, or we can set to 0 to simplify:

$$S_{summer} = 1 - M_{summer}. \tag{8}$$

The Beverton-Holt function is again used for including carrying capacity, $K$:

$$\pi_{summer} = \frac{1}{\left(\frac{1}{S_{summer}} + \frac{n_{spring.larvae}}{K_{summer}}\right)}, \tag{9}$$

where, like in [equation 5](), $S_{summer}$ is the baseline summer larval survival, without density dependence (defined by the user via summer mortality in [equation 8]()). Larval lamprey surviving the summer are then drawn from a binomial distribution, using $\pi_{summer}$ as the survival probability:

$$n_{fall.larvae} \sim \text{ binomial} \left(n_{spring.larvae}, \pi_{summer}\right) \tag{10}$$

Winter conditions are explored analogously to summer conditions. Similar to summer conditions, winter conditions can be explored by setting the winter rearing capacity, $K_{winter.larvae}$ and winter mortality, $M_{winter}$. Again, this affects all age-1+larvae together, regardless of size. The equations for winter conditions are the same as for summer conditions:

$$S_{winter} = 1 - M_{winter} \tag{11}$$

$$\pi_{winter} = \frac{1}{\left(\frac{1}{S_{winter}} + \frac{n_{fall.larvae}}{K_{winter}}\right)} \tag{12}$$

$$n_{spring+1.larvae} \sim \text{ binomial} \left(n_{fall.larvae}, \pi_{winter}\right) \tag{13}$$

This process is used to calculate the density-dependent survival, $\pi_{dd}$, which is applied to each age class, along with an age-class specific density-independent survival, $\pi_{di_l}$, which is specified by the user.

$$\pi_{dd} = \frac{n_{spring+1.larvae}}{n_{spring.larvae}} \tag{14}$$

Surviving larvae in each age-class, $n_{survived.larvae_l}$, is drawn from the pool of available larvae in that age-class, $n_{larvae_l}$:

$$n_{survived.larvae_l} \sim \text{ binomial} \left(n_{larvae_l}, \pi_{dd} \cdot \pi_{di_l}\right). \tag{15}$$

**Larval age classes and transformation.** Each year, surviving eye-less larvae (also referred to as ammocoetes; [24]) either transform into eyed juveniles (also referred to as macrophthalmia; [24]) and head to the ocean or remain in fresh water and advance to the next larval age-class. This transformation stage can take many months ([Fig 1]()); thus, we have identified the intermediate "transformer" stage in the following math.

Any surviving age-2 to age-10 larvae in the model can transform with probability $\delta_l$ for age class $l$. Individuals are stochastically drawn from a binomial distribution:

$$n_{transform_l} \sim \text{ binomial} \left(n_{survived.larvae_l}, \delta_l\right) \tag{16}$$

The total number of transformers is the sum of individuals that transformed from each age class, $l$:

$$n_{transformers} = \sum_{l=2}^{10} n_{transform_l}, \tag{17}$$

once larval fish reach 10 years of age, they all transform ($\delta_{10} = 1$).

Larvae that do not enter the intermediate transformer life stage will enter the next age class with probability $(1 - \delta_l)$, where $(1 - \delta_l)$ is the probability of not transforming in a given year. The next age class of larvae could then be, again, drawn from a binomial distribution:

$$n_{larvae_{l+1}} \sim \text{ binomial} \left(n_{survived.larvae_l}, 1 - \delta_l\right) \tag{18}$$

However, fish that have already transformed are removed from the pool of larvae that can move to the next age class. Thus, to ensure that larval lamprey are not being created nor destroyed (via stochastic resampling of the same individuals), we instead subtract the number of transformed fish from the number of surviving larvae (within that age class):

$$n_{larvae_{l+1}} = n_{survived.larvae_l} - n_{transform_l} \tag{19}$$

**Downstream barriers and juveniles at sea.** Before juvenile lamprey reach the ocean, they encounter any downstream barriers. The user can specify any number of barriers and the probability of passage for each barrier (or average proportion of fish that pass those barriers). The resulting number of juveniles that continue towards the marine environment, $n_{juveniles_M}$, (soon to become ocean age-0 juveniles once they survive ocean entry) is estimated as:

$$n_{juveniles_M} \sim \text{binomial}\,(n_{transformers}, \theta_{DS}) \tag{20}$$

where $\theta_{DS}$ is the total downstream probability of riverine juveniles, $n_{juveniles}$, passing all barriers. The total downstream probability of passing barriers equals the product of all the probabilities of passing each successive downstream barrier, $\theta_{D_d}$, for $d$ downstream barriers:

$$\theta_{DS} \;=\; \prod_{k\,=\,1}^{d} \theta_{D_d}. \tag{21}$$

If there are no barriers, $d = 0$, $\theta_{DS} = 1$, and $n_{juveniles_M} = n_{transformers}$. However, if the translocation option is selected, translocated juveniles ($n_{juveniles_T}$) are removed from the fish experiencing barrier passage, and instead are moved below the barriers, with the user-defined juvenile translocation survival probability $\theta_{JT}$:

$$n_{juveniles_P} \sim \text{binomial}\,(n_{juveniles} - n_{juveniles_T}, \theta_{DS})\,, \tag{22}$$

$$n_{juveniles_{ST}} \sim \text{binomial}\,(n_{juveniles_T}, \theta_{JT})\,. \tag{23}$$

Fish that experienced and passed barriers ($n_{juveniles_P}$) are then summed with surviving translocated fish ($n_{juveniles_{ST}}$), to calculate the number of juveniles entering the ocean, in lieu of equation 20:

$$n_{juveniles_M} = n_{juveniles_P} + n_{juveniles_{ST}} \tag{24}$$

Juveniles then may experience mortality as they meet marine predators in the ocean (e.g., the river plume). We again use moment matching [37], to include our moments (mean, μ and standard deviation σ of ocean entry survival, $\pi_{entry.ocean}$) in the beta distribution parameters, $\alpha$ and $\beta$:

$$\alpha = \left( \frac{\mu \cdot (1-\mu)}{\sigma_{EO}^2} - 1 \right) \cdot \mu, \tag{25}$$

$$\beta = \left( \frac{\mu \cdot (1-\mu)}{\sigma_{EO}^2} - 1 \right) \cdot (1-\mu), \tag{26}$$

$$\pi_{entry.ocean} \sim \text{beta}(\alpha, \beta). \tag{27}$$

We can then draw our resulting juvenile fish (in the 0$^{th}$ juvenile age-class) from the number of juveniles entering the marine environment with probability $\pi_{\text{entry.ocean}}$:

$$n_{juvenile_0} \sim \text{ binomial} \left( n_{juveniles_M}, \pi_{\text{entry.ocean}} \right).$$
(28)

**Juveniles at sea to adults in river.** Once juveniles are at sea, each age class, $j$, can transition to an adult returner, heading upstream with an age-class specific probability $\zeta_j$, and have a river entry survival probability $\pi_{\text{entry.river}}$. Both $\zeta_j$ and $\pi_{\text{entry.river}}$ are ultimately defined by the user, however $\pi_{\text{entry.river}}$ is specified with a mean, μ, and standard deviation, σ, for river entry survival as in equations 25–27:

$$n_{return_j} \sim \text{ binomial} \left( n_{juvenile_j}, \zeta_j \cdot \pi_{\text{entry.river}} \right),$$
(29)

where $n_{return_j}$ is the number of juveniles in class, $j$, that will return to a riverine system. The sum across all age classes is the total in-river adults ($n_{adults_R}$):

$$n_{adults_R} = \sum_{j=1}^{10} n_{return_j}$$
(30)

Juveniles that remain at sea advance to the next juvenile age-class with a probability equal to yearly ocean survival (which is parameterized with different user-defined inputs, but relies on the same equations as the ocean entry survival above):

$$n_{juvenile_{j+1}} \sim \text{ binomial} \left( n_{juvenile_j} - n_{return_j}, \pi_{\text{yearly.ocean}} \right).$$
(31)

That is, $\pi_{\text{yearly.ocean}}$ is specified with a mean, μ, and standard deviation, σ, for yearly ocean survival as in equations 25–27. If fish spend 10 years as juveniles, they automatically swim upriver if they do not happen to die during the process.

**Upstream barriers and translocation.** Adults swimming upstream may encounter upstream barriers, which impede the probability of fish passage. The resulting number of adult fish of age-class 0 in the river, $n_{adult_0}$, is estimated as:

$$n_{adult_0} \sim \text{ binomial} \left( n_{adults_R}, \theta_{\text{US}} \right)$$
(32)

where $\theta_{\text{US}}$ is the total upstream probability of each adult, $n_{adults_R}$, passing all barriers. The total upstream passability, $\theta_{\text{US}}$, equals the product of all the probabilities of passing each successive barrier, $\theta_{U_u}$, for $u$ upstream barriers:

$$\theta_{\text{US}} = \prod_{k=1}^{u} \theta_{U_u}$$
(33)

However, in the event that the translocation option is selected, translocated adults ($n_{adult_T}$) are removed from the fish experiencing barrier passage, and instead are dropped off above the barriers, with the user-defined adult translocation survival probability $\theta_{AT}$:

$$n_{adult_P} \sim \text{ binomial} \left( n_{adults_R} - n_{adult_T}, \theta_{\text{US}} \right),$$
(34)

$$n_{adult_{ST}} \sim \text{ binomial} \left( n_{adult_T}, \theta_{AT} \right).$$
(35)

Fish that experienced and passed barriers ($n_{adult_P}$) are then summed with surviving translocated fish ($n_{adult_{ST}}$):

$$n_{adult_0} = n_{adult_P} + n_{adult_{ST}}$$
(36)

**Adults to spawners.** Some adults may spawn immediately (adult age-0, $n_{adult_0}$), while others will hold in the river without feeding and spawn within the next two years (adult ages 1 and 2). Adult age-classes, $a$, spawn in a given year with a probability $\eta_a$, where age-2 adults all spawn ($\eta_2=1$):

$$n_{spawn_a} \sim \text{binomial}\left(n_{adult_a}, \eta_a\right). \tag{37}$$

The sum across all age classes is the total spawners for that year:

$$n_{spawners} = \sum_{a=0}^{2} n_{spawn_a} \tag{38}$$

Spawners now contribute to the next generation via equations 1 and 3 and the life cycle continues to the next year.

### Default parameter values

Pacific lamprey occupy a broad geographic range [3]. Thus, many of their life cycle parameters are likely to vary across their range. Here, we provide the model structure and some rationale for default lamprey life cycle parameters. That is, if the user runs the model code in R, or in the Shiny app, without specifying parameters, the defaults will be automatically used. However, it is up to the user to specify the parameters that most closely represents their system and lamprey species of interest. Sometimes information is limited, and one will have to make assumptions about how to use information from other populations or species. Models are dependent on information to describe interactions, relationships, and other processes, and process-based models, in particular, can incorporate a broad range of information types [42]. While empirical data are often used to parameterize key model processes, these mathematical models are flexible such that gaps in knowledge can be represented by ecological theory, expert opinion, educated guesses, or informed by sensitivity analyses (see the *Global sensitivity analysis* section below) that explore the entire range of realistic possibilities. We provide a description of default parameters and where they come from (Table 1) but realize that these parameters may not be appropriate for every population of Pacific lamprey or other lamprey species.

**Spawning and fecundity.** An initial number of spawners is a necessary simulation input. After the first year the internal population dynamics determine the spawner numbers in subsequent years. An arbitrary default of 800 spawners was selected. Longer burn-in time (also known as warm-up or simulation years that are thrown away [43]) will ensure that this initial number of spawners does not affect the simulation since internal dynamics will dominate after 30–40 years (Fig 3). The number of females is determined each year by the female sex ratio, which is defaulted to 0.5 (or 50% females). This assumption might not always be appropriate. In some fishes, for example, the sex ratio can be determined by the environment, including temperature and density [4,44]. In sea lamprey, specifically, the sex ratio can be determined by larval growth rates [45]. Currently, users will need to set the sex ratio within the LCCM to one fixed value for the duration of a simulation. However, alternative scenarios can explore the role that different sex ratios play in population dynamics. Individual female fecundity is drawn from a distribution (equation 1) with a mean, $\lambda_{fecund}$, and standard deviation, $\sigma_{fecund}$, defaulted to 127,000 and 33,500 eggs, respectively (rounded to the nearest 500 eggs [34]).

**Egg survival.** Work with other lamprey species suggests that about 12% of eggs stay in the redd and are successfully hatched, while 81.5% are washed out of the redd, and the remainder are unfertilized or not viable for other reasons [46,47]. Of the 81.5% of washed-out eggs, we assume that a survival rate around 22% is reasonable (Ralph Lampman, viability of eggs in various media; [48]), suggesting another ~18% of total eggs survive ($0.815 \times 0.22 \approx 0.18$). Combined, we estimate about 30% of eggs hatch (12% + 18%). Of these, we estimate that only about 6% of these individuals survive until first-feeding larval stage (Ralph Lampman, natural nutrient level hatchery setting; [49,50]), which means total egg to age-0 larvae survival, $S_{egg}$, is roughly 2% ($0.3 \times 0.06 = 0.018 \approx 0.02$).

**Table 1. User-defined parameters and default values with references. All references are for Pacific lamprey (*Entosphenus tridentatus*), unless otherwise stated in the column, "Reference River Basin." Additionally, the Manion & Hanson, 1980 reference includes information on sea lamprey (*Petromyzon marinus*), silver lamprey (*Ichthyomyzon unicuspis*), chestnut lamprey (*I. castaneus*), northern brook lamprey (*I. fossor*), and American brook lamprey (*Lethenteron appendix*), but not Pacific lamprey.**

| Parameter | Description | Default | Default Reference | Reference River Basin |
|---|---|---|---|---|
| $\lambda_{fecund}$ | Mean fecundity. | 127000 | Clemens et al., 2013 | Willamette River Basin (OR), Klamath River estuary (CA) |
| $\sigma_{fecund}$ | Standard deviation of fecundity. | 33500 | Clemens et al., 2013 | Willamette River Basin (OR), Klamath River estuary (CA) |
| $n_{spawners}$ | Initial spawner abundance. After initialization, population dynamics determines spawner numbers (see Fig 3). | 800 | – | – |
| ♀ : ♂ | Female sex ratio. | 0.5 | Assumed | – |
| $S_{egg}$ | Egg to larva survival, without density dependence. Used with carrying capacity, $K_{egg}$, to calculate density-dependent survival, $\pi_{egg}$ (equation 5). | 0.02 | Manion & Hanson, 1980; Hardisty 2006; Lampman et al. 2016; 2021 | Multiple other lamprey species (see caption) |
| $K_{egg}$ | Carrying capacity of eggs in the system or spawning capacity (in # eggs). Alternatively, egg density and stream area can be provided to calculate $K$. Used with egg survival, $S_{egg}$, to calculate density-dependent survival, $\pi_{egg}$ (equation 5). | 10 million, 575 eggs/m$^2$ | Schultz et al., 2014 | Willamette River Basin (OR) |
| $n_{hatch.larvae_l}$ $n_{hatch.transformer}$ | The user can select any number of hatchery releases consisting of any number of age-1 to age-10 larvae and transformers. | 0 | – | – |
| $M_{summer}$ $M_{winter}$ | Additional larval mortality in summer/ winter. Used to calculate density-dependent survival, $\pi_{dd}$ (see equations 8–14). | 0 | – | – |
| $K_{summer}$ $K_{winter}$ | Carrying capacity of larvae in the system in summer/winter (in # larvae). Alternatively, larval density and stream area can be provided to calculate $K$. Used to calculate density-dependent survival, $\pi_{dd}$ (see equations 9 and 12). | 1 million, or 0.13 larvae/m$^2$ | Schultz et al., 2014 | Willamette River Basin (OR) |
| $\pi_{di_l}$ | The density-independent survival probability of larvae, in each age-class $I$. A value is specified for each age-class (0–10) or one value for all age-classes can be input. | 0.33, 0.45, 0.61, 0.69, 0.74, 0.77, 0.79, 0.8, 0.8, 0.8, 0.8 | Schultz et al., 2014 | Willamette River Basin (OR) |
| $\delta_l$ | The transformation probability of larvae, in each age-class $I$. A value is specified for each age-class or one value for all age-classes can be input. | 0, 0, 0.002, 0.042, 0.555, 1, 1, 1, 1, 1, 1 | Lampman et al., 2020 | Yakima River Basin (WA) |
| $d$ | The number of downstream barriers. | 0 | – | – |
| $\theta_{D_d}$ | The probabilities of passing each successive downstream barrier for $d$ downstream barriers. Downstream translocation survival below. | 0.9 | Deng et al., 2023 | Lower Granite Dam (Columbia River) |
| $\mu_{entry.ocean}$ | Mean survival as juveniles first enter the ocean via the river mouth. | 0.46 | Brosnan et al., 2014 | Chinook salmon (*Oncorhynchus tshawytscha*) in lower Columbia River |
| $\sigma_{entry.ocean}$ | Survival standard deviation as juveniles first enter the ocean. | 0.09 | | |
| $\zeta_j$ | Probability that each juvenile age class, $j$, heads upstream (ocean years 0–10). | 0, 0.02, 0.05, 0.1, 0.15, 0.45, 1, 1, 1, 1, 1 | Hess et al., 2022 | Snake River |
| $\mu_{yearly.ocean}$ | Mean yearly survival for juvenile lamprey that remain at sea. | 0.7 | Lampman et al., 2015 | Yakima River Basin (WA) |
| $\sigma_{yearly.ocean}$ | Yearly survival standard deviation for lamprey that remain at sea. | 0.1 | | |

*(Continued)*

**Table 1.** (Continued)

| Parameter | Description | Default | Default Reference | Reference River Basin |
|---|---|---|---|---|
| $\mu_{entry.river}$ | Mean survival as lamprey enter the river mouth. | 0.67 | Wargo Rub et al., 2019 | Chinook salmon in lower Columbia River |
| $\sigma_{entry.river}$ | Survival standard deviation as lamprey enter the river mouth. | 0.09 | | |
| $u$ | The number of upstream barriers. | 0 | – | – |
| $\theta_{U_u}$ | The probabilities of passing each successive upstream barrier for $u$ downstream barriers. | 0.6 | Moser et al., 2002; Keefer et al., 2013 | Bonneville, The Dalles, and McNary Dams (Columbia River) |
| $n_{transformers_T}$ $n_{adult_T}$ | The number of translocated transformers and adults (to be translocated around all downstream and upstream barriers). | 0 | – | – |
| $\theta_{T_t}$ $\theta_{T_a}$ | The survival probability of translocation for transformers and adults. | 0.99 | Aaron Jackson, *personal communication* | Columbia River |
| $\eta_a$ | The probability of individuals in each adult age-class, $a$, spawning in a given year (0–3). Individuals can remain in stream for up to 2 years before spawning ($\eta_2$=1). | 0.05, 0.7, 1 | Ralph Lampman, *unpublished* data; Aaron Jackson, *personal communication* | – |

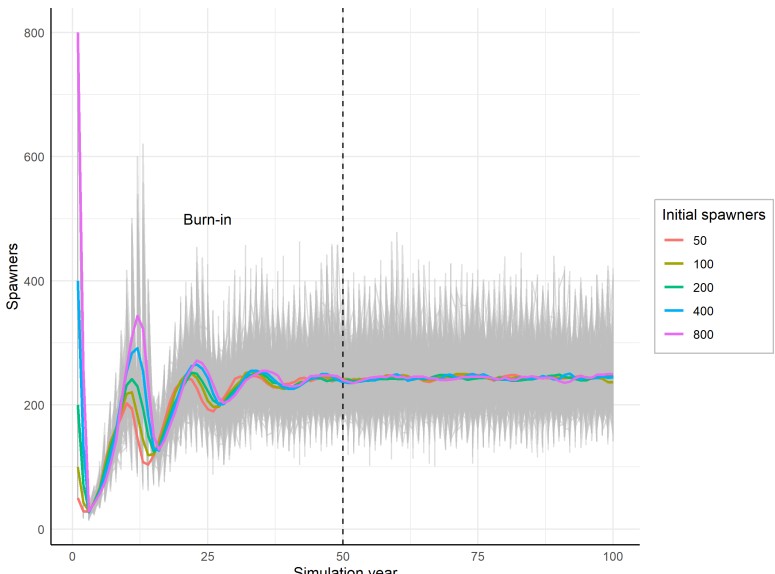

**Fig 3. Initialization of the model with 5 different starting values for initial spawners leads to similar results after a conservative burn-in period of 50 years.** Colors indicate mean values for 100 stochastic runs of each initial starting value for spawners, while grey lines represent variation (uncertainty) in stochastic simulation runs. It is important to note that the absolute number of spawners may not be meaningful outside the context of the particular values selected for model life cycle parameters. For example, if arbitrary parameters (such as larval survival or carrying capacity values) are selected that do not represent any particular stream, the absolute numbers in the model output may not necessarily be biologically meaningful. Yet, even in these cases where absolute numbers are not calibrated with empirical data, relative differences amongst scenarios can be informative as we compare the effects of different processes.

Total egg carrying capacity, $K_{egg}$, can be input directly by the user (default is 575 million eggs, see below), or it can be calculated from the river area and mean egg density measured at relevant sites. The river area can be calculated by multiplying the total river length (km) of available habitat by the mean width of suitable habitat (also in km). The model allows

calculation of the mean carrying capacity across the range of heterogeneous locations (likely in real-world conditions). We used previous work (Table 3-1 in [51]) to calculate an average default egg density of 575 eggs/m$^2$. To do this we divided the mean peak redd density (redds/km) by the mean stream width (converted from m to km) to get areal redd density (redds/km$^2$) and then multiplied this by the default fecundity (127,000 eggs see above) to get eggs/km$^2$. If egg carrying capacity is defined, instead of egg density, the default of 575 million eggs is identical to using the default density of 575 eggs/m$^2$ with 1 km$^2$ of available spawning habitat.

**Larval survival.** Total summer and winter larval carrying capacity, $K_{summer}$ and $K_{winter}$, can be input directly by the user (default is 1.3 million larvae during each time period, see below), or it can be calculated from the total river area and mean larval density measured at relevant sites. The default mean density (individuals/m$^2$) for larvae is 0.13, which is taken as an average across 14 tributaries from the Willamette River Basin (Appendix 1 in [51]) and 10 tributaries of the Columbia River Basin (Table 4 found in [52]). If larval carrying capacity is defined, instead of larval density, the default of 1.3 million larvae is identical to using the default density of 0.13 larvae/m$^2$ with 10 km$^2$ of available rearing habitat (mean density of 0.13 individuals/m$^2$, multiplied by conversion of m$^2$ to km$^2$ [1 million], multiplied by an arbitrary default of 10 km$^2$ rearing habitat).

Survival probability can be selected by age class (age-0 – age-10; default) or with the same probability for each age class (less realistic but requires fewer decisions). Default larval survival by age-class values were taken from Schultz et al., 2014 (Table 5-1 found in [51], Chen/Watanabe estimates without age-0, since age-0 survival is input directly here). Only values for age-1 to age-7 were provided by Schultz et al., 2014 [51]. We estimated age-0 survival to be 33% because a 0.33 age-0 survival makes cumulative survival across age-0 – age-7 to be 22.5%, which is in-line with previous estimates of 19–26% for entire larval life cycle survival [51]. To estimate age-8 through age-10 survival, we fit a sigmoid curve to age-1 to age-7 data [51] with R package `nls` to predict the survival values, which were rounded to the nearest 2 decimal places (see Table 1; [23]).

**Larval transformation by age-class.** Transformation probability can be selected by age class (age-0 – age-10; default) or with the same probability for each age class (less realistic but requires fewer decisions). Transformation probabilities were modified until matched empirical data for the Yakima River Basin (age-2: 1%, age-3: 9%, age-4: 62%, and age-5: 28%, R. Lampman, [53]). This particular set of values (see Table 1) only matches the Yakima River empirical transformer composition data given the other default model parameters. Thus, if users change life cycle parameters, such as egg or larval survival, these transformation probabilities will need to be modified again to match the desired transformer composition. These particular values also do not allow larvae to reach age-6, as they will automatically transform at age-5, assuming they survive. This is realistic for some streams, but not others. These values should be adjusted to fit the age-class distribution for the stream of interest. That is, the user can adjust transformation probabilities such that transformer age class composition matches that of the focal system.

**Entering the marine environment.** We used a value of 0.9 (90%) for the default downstream passage success per dam, $\theta_{D_a}$. This value is estimated at Lower Granite Dam [41]. To our knowledge, data do not exist specifically for ocean entry survival (Laurie Weitkamp, *personal communication*). We assume that marine predators in the plume are opportunistically consuming available prey. Thus, as a rough approximation, we have taken ocean entry survival as the mean (across years) value of Columbia River plume survival for Chinook salmon (*Oncorhynchus tshawytscha*) smolts (survival±SD: 0.46±0.09 from Table 1 in [54]). Annual survival for juveniles in the ocean (referred to here as yearly ocean survival) is assumed to be substantially higher, and we used an assumed value of 0.7±0.1 (mean±SD), because it led to reasonable total lifetime survival values.

**Re-entering freshwater and spawning.** The probability that each juvenile age class (0–10) heads upstream, $\zeta_j$, is set to 0, 0.02, 0.05, 0.1, 0.15, 0.45, 1, 1, 1, 1, 1 as a default. These values are based on age-class distributions for each life stage of Snake River Pacific lamprey [30]. Fig 5 in Hess et al., (2022) [30] suggests that most fish spend 5 or 6 years in the ocean; thus, we have selected nearly half (0.45) of 5$^{th}$ ocean year juveniles and all of the remaining ($\zeta_6 = 1$) 6$^{th}$ ocean year juveniles to swim upriver. With this parameterization fish never spend more than 6 years in the ocean [30], but this can be extended by the user ($\zeta_6 < 1$).

Once lamprey enter the mouth of the river, they may experience increased levels of predation or mortality at these physical bottlenecks [55], which is controlled in this model via a river entry survival. Given that predation on lamprey re-entering riverine systems (such as by pinnipeds) can be equal to, or higher than, predation on salmonids [55,56], we have used information on river mouth survival of adult salmonids as a stand-in for the more data-limited lamprey populations. Thus, default survival for Pacific lamprey entering the river mouth was taken as $0.67 \pm 0.09$ (mean $\pm$ SD; from Fig 5 in [57]; [23]).

For fish encountering barriers, we used a value of 0.6 (60%) for the default upstream passage success, $\theta_{U_u}$ (the mean of all lower and upper extreme values estimated at Bonneville [38–47%], The Dalles [50–82%], and McNary [65–75%] dams [58,59]). Any number of available adult fish can be translocated above all barriers [30], but the user must specify the survival probability of translocation. We use a default value of 0.99 for adult translocation survival (Aaron Jackson, *personal communication*).

Adults can wait in the river for up to two years prior to spawning. As a default probability of spawning for entry-year adults (adult age-class 0) and adults that have waited one year (adult age-class 1), we have used 0.05 and 0.7, respectively, to reflect that most Pacific lamprey appear to wait at least one year, while we assume all adult age-class 2 individuals spawn after two years ($\eta_2$=1).

## Global sensitivity analysis

Sensitivity analyses allow for an exploration of model sensitivities to particular parameters. As opposed to altering one parameter at a time, and keeping all others at fixed values, global sensitivity analyses (GSA) allow an assessment of parameter influence on overall model dynamics while considering all possible interactions among parameters [60,61]. A GSA is used to evaluate how variations in input parameters influence the uncertainty in the output of the model, which helps identify the inputs that have the most significant impact, assess interactions among variables, and determine whether certain inputs jointly affect the results. By understanding these sensitivities, GSA improves model robustness, ensuring it is not overly sensitive to small changes that could indicate instability. It also guides data collection efforts by prioritizing key parameters that significantly affect predictions. Additionally, GSA supports decision-making in areas like risk assessment, optimization of management strategies, and policy-making by clarifying how different inputs drive outcomes. Furthermore, GSA aids in validating model assumptions by verifying whether simplifications or approximations are justified based on the sensitivity of inputs. Unlike local sensitivity analysis, which examines small perturbations around a fixed point, GSA evaluates input variations across their entire range, making GSA a more comprehensive approach for analyzing complex systems. A GSA is advantageous to fixing all non-modified parameter values to mean values and assuming that they are correct, because parameter space, and thus parameter influence, is more thoroughly explored.

To more efficiently sample parameter space, we used Latin hypercube sampling (LHS). LHS generates near-random sampling of parameter values from a multidimensional distribution where the sampling space is broken up into strata or breaks [62,63]. Each break in the sampling space is randomly sampled only once for each parameter such that the range of parameter values is efficiently sampled across all parameters simultaneously. We used the R [21] package `lhs` [64] to set up two global sensitivity analysis where parameter values were sampled with a Latin hypercube with 100 breaks.

For the first GSA, we sampled parameters more broadly. For example, values that are constrained between 0 and 1 (e.g., survival, transformation probabilities, etc.) are mostly not empirically resolved for Pacific lamprey and, thus, were sampled across the entire range of possible values (0–1) to fully explore potential influence of the parameter space. Fecundity is hypothetically only constrained on the lower end to 0, whereas the upper end could be infinite. Since this was not feasible, fecundity was explored with a normal distribution with the mean and standard deviation as defined in *Default values* (see Tables 1 and 2). Carrying capacities vary by location and spatial extent; thus, they were explored across a large range of values with uniform distributions:

$$K_{egg} \sim uniform\left(1 \times 10^5, 1 \times 10^{12}\right) \tag{39}$$

$$K_{summer} \sim uniform\left(1 \times 10^4, 1 \times 10^8\right) \tag{40}$$

$$K_{winter} \sim uniform\left(1 \times 10^4, 1 \times 10^8\right) \tag{41}$$

Since the GSA relies on LHS, the deterministic option (function argument) was turned on in the `MODEL.R` code, and, thus, standard deviation values are not necessary to systematically sample (i.e., the LHS handles the variation in inputs already).

For each of the 100 GSA runs (100 LHS breaks), we ran the model for 50 years after 50 additional burn-in years (Fig 3). Since stochastic sampling was turned off, each run reached nearly constant stable states after a conservative 50-year burn-in. The resulting number of spawners in the last year of the simulation was used as the response variable in a random forest model to assess parameter influence.

We used a random forest regression model with 501 trees to assess the relationship between the number of spawning lamprey (response) to the 44 varying parameter values in the GSA. Using `tuneRF` in the `randomForest` package [65], we determined 8 variables were a near-optimal number of variables to try for each tree split. We then used the function `importance` to determine the importance of each of the 44 variables, as measured by the percent increase in mean squared error (MSE) when each variable is included. The MSE on the out-of-bag (OOB) portion of the data (data not included in training the model) is calculated for each tree, then the same is done after permuting each variable in the model. The difference in MSE values between the initial regression forest and the permutation of each variable are averaged across all trees and normalized by the standard deviation of the differences to generate a percent change in MSE when a given variable was permuted. A higher percent increase in MSE suggests that a randomly permuted variable is more important/influential than another randomly permuted variable that leads to lower changes is MSE. To visualize uncertainty in importance metrics, we ran 100 random forest models and plotted the mean and 95% confidence intervals (as 1.96 x SE; Gomes, 2024; Figs 4 and 5).

The second GSA was conducted similarly to the first, except that the parameter space was constrained to more reasonable values (see Table 2). For example, instead of sampling uniformly between 0 and 1 for the sex ratio, this parameter-restricted GSA explored sex ratios drawn from a beta distribution with mean of 0.5 (50% males and females) and a standard deviation of 0.05. That is,

$$[\female : \male] \sim beta(\alpha, \beta), \tag{42}$$

where $\alpha$ and $\beta$ are moment matched to $\mu$ and $\sigma$ via equations 25 and 26 [37]. All of the remaining parameters were similarly drawn from a beta distribution with their respective mean and standard deviations. All parameters retained their mean values from the *Default parameters* section (Table 1) except where probabilities were too close to 0 or 1 for proper sampling (Table 2). In these cases, values of 0.05 and 0.9, respectively, were used. Resulting beta distributions were visualized to ensure that values of 0 and 1 were well-represented. Most parameters were sampled with a conservative standard deviation equal to 0.1, to ensure that a broad (yet more restricted relative to the first GSA) parameter space was sampled (see Table 2).

**Management scenario case study examples**

Management scenarios can be created and combined in many ways for lamprey conservation and or control. For example, to explore applications of lampricides to kill larvae in the case of control of invasive sea lamprey, one might choose to alter larval survival in a number of different scenarios. Alternatively, model users might explore increasing larval survival as improvements to rearing habitat quality. Thus, users can have different, but valid, rationales for exploring and forcing

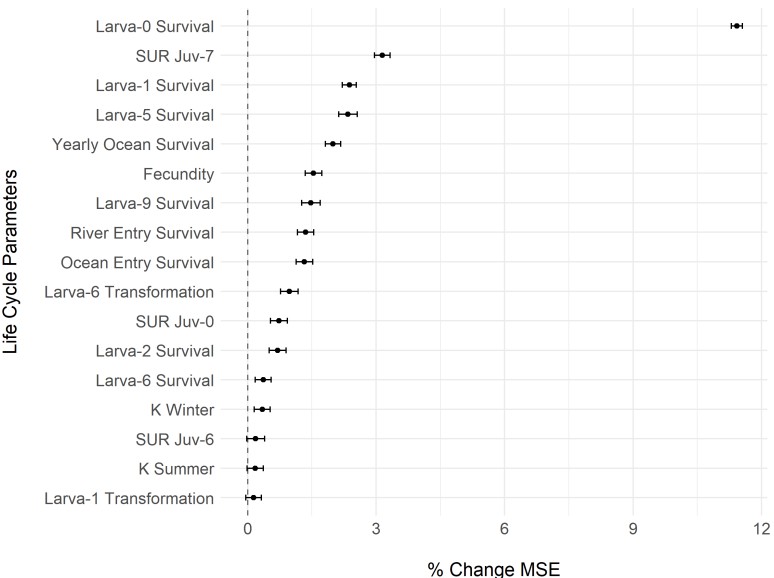

**Fig 4. Broad global sensitivity analysis (GSA) with parameter space sampled across a full range of possible values (see Table 2).** Importance of life cycle parameters in predicting the number of spawners (% change in mean squared error, MSE). We used a global sensitivity analysis (GSA) and a Latin hypercube to systematically vary all life cycle model parameters simultaneously across a broad range of possible values. We then ran the deterministic versions of the life cycle model for 100 years with 100 different parameter sets. With the resulting model output for the number of spawners, 100 random forests, each with 501 trees, were used to regress the number of spawners across the 100 life cycle parameter sets for the 44 life cycle variables (only those with positive % change in MSE are shown on the y-axis here, others are omitted for brevity). Dots indicate mean parameter importance and error bars indicate 95% confidence intervals (as 1.96 x SE). Positive x-axis values indicate an improvement in model skill (when y-axis parameter is not randomly permuted) and negative values indicate a reduction in model skill (relative to random permutations). K = carrying capacity; SUR = swim upriver probability for each juvenile ocean age-class.

changes to the same parameter values. To demonstrate potential management applications, we chose five simple scenarios to explore (Table 3).

We selected these five scenarios to broadly represent common management actions relating to Pacific lamprey conservation [66], including the influence of barriers [67–69], translocations [30,70], and hatchery releases [49,50] on lamprey populations. Each scenario consists of 100 stochastic replications of a 100-year simulation (50-year run time + 50-year burn-in period). The first scenario is considered baseline conditions, where only default parameter values are input. The second scenario adds one upstream barrier with the default 60% passage probability. In this scenario, the number of adults immigrating to freshwater are reduced by 60%. The third scenario is the same as the second but includes the translocation of up to 100 adults for each year, within each simulation. If there are fewer adults available in the model in any given year, then all adults are translocated above the barrier for that year. The fourth scenario is also the same as the second but includes yearly hatchery releases of 1000 fish at the 'transformer' stage, which is the short stage in between larvae and juveniles, before the fish have made it to the ocean (Figs 1 and 2). The fifth scenario is a combination of scenarios 3 and 4, which includes one upstream barrier with the default 60% passage probability, the translocation of up to 100 adults for each year, and yearly hatchery releases of 1000 fish at the 'transformer' stage. Results are visualized as the median value of the 100 replications for each of the 5 scenarios for each year (see Fig 6 and function `PlotTSComparison` in [23]).

## Results

Simulations with default parameter values, during 100-year simulations, were stable and did not lead to population extinctions or explosions. The initial number of spawners did not affect the final number of spawners after a sufficient burn-in

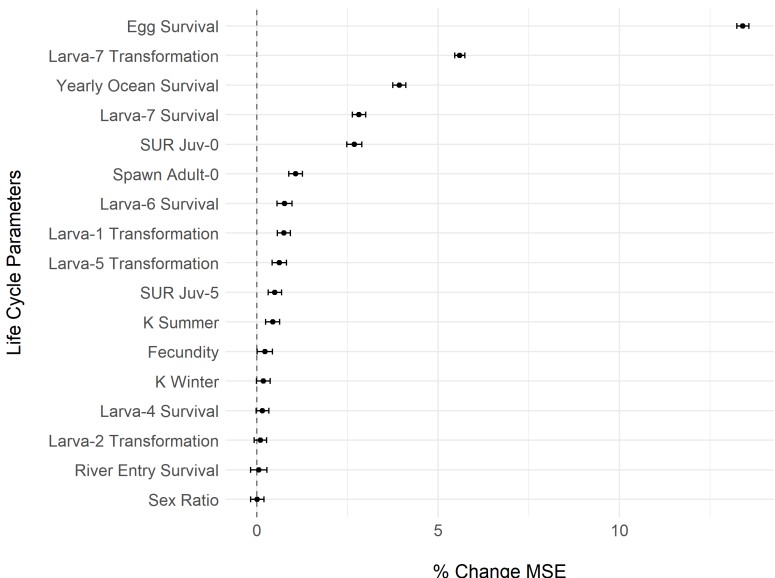

**Fig 5. Restricted global sensitivity analysis (GSA) with parameter space sampled across more reasonable values (see Table 2).** Importance of life cycle parameters in predicting the number of spawners (% change in mean squared error, MSE). We used a global sensitivity analysis (GSA) and a Latin hypercube to systematically vary all life cycle model parameters across a range of reasonable values. We then ran the deterministic versions of the life cycle model for 100 years with 100 different parameter sets. With the resulting model output for the number of spawners, 100 random forests, each with 501 trees, were used to regress the number of spawners across the 100 life cycle parameter sets for the 44 life cycle variables (only those with positive % change in MSE are shown on the y-axis here, others are omitted for brevity). Dots indicate mean parameter importance and error bars indicate 95% confidence intervals (as 1.96 x SE). Positive x-axis values indicate an improvement in model skill (when y-axis parameter is not randomly permuted) and negative values indicate a reduction in model skill (relative to random permutations). K = carrying capacity; SUR = swim upriver probability for each juvenile ocean age-class.

period (Fig 3). The burn-in period smoothed the initial startup population dynamics and leads to more consistent results across minor changes in initial conditions.

## Global sensitivity analysis

The global sensitivity analysis (GSA) with the broad parameter sampling scheme suggested that the most sensitive parameters in the model were age-0 larval survival, juvenile ocean age-7 swim upriver probability, and age-1 and age-5 larval survival (Fig 4). When more realistic parameter values were explored in a second GSA, the most sensitive parameters in the model were egg survival, age-7 larval transformation probability, yearly ocean survival, and age-7 larval survival (Fig 5). In both GSA random forest analyses, 17 of 44 parameters had positive effects (i.e., they were considered influential) on predicting (measured as % mean squared error) the number of spawners at the end of a 100-year simulation. Many of these influential parameters were the same across the two analyses (yearly ocean survival, fecundity, river entry survival, summer and winter larval carrying capacity [K], age-6 larval survival, age-1 larval transformation probability), or very nearly the same (multiple ages of larval survival and transformation probabilities, and juvenile ocean age-5, -6, and -7 probabilities of swimming upriver), yet their order of importance differed between the two analyses.

## Management scenario sensitivity analysis

The "Baseline" conditions in scenario 1 reached a mostly stable spawning population around 200 individuals. Scenario 2, the addition of an upstream barrier with 60% passability, led to a drop in the spawning population (relative to baseline conditions, Fig 6). In both scenarios 3 and 4, the number of spawners made up roughly half of the consequences of adding an

**Table 2. Distributions and hyperparameters for global sensitivity analyses (GSA). The broad GSA explored a larger parameter space, relying on a uniform distribution across the entire range of possible values (0–1 in most cases). The restricted GSA explored a more constrained, and realistic, parameter space.**

| Parameter | Description | Broad GSA | Restricted GSA |
|---|---|---|---|
| – | Years | 50 (+50 burn-in) | 50 (+50 burn-in) |
| $\lambda_{fecund}$ | Fecundity | Normal ($\mu = 127000$, $\sigma = 33500$) | Normal ($\mu = 127000$, $\sigma = 33500$) |
| ♀ : ♂ | Sex ratio | Uniform (0, 1) | Beta ($\alpha, \beta$), $\mu = 0.5$, $\sigma = 0.05$ |
| $S_{egg}$ | Egg survival | Uniform (0, 1) | Beta ($\alpha, \beta$), $\mu = 0.02$, $\sigma = 0.04$ |
| $K_{egg}$ | Egg carrying capacity | Uniform ($1 \times 10^5$, $1 \times 10^{12}$) | Uniform ($1 \times 10^5$, $1 \times 10^{12}$) |
| $K_{summer}$ $K_{winter}$ | Summer and winter carrying capacity | Uniform ($1 \times 10^4$, $1 \times 10^8$) | Uniform ($1 \times 10^4$, $1 \times 10^8$) |
| $\pi_{di_l}$ | Larval survival for ages 0–10 | Uniform (0, 1) | Beta ($\alpha, \beta$) $\mu_l$ = 0.33, 0.45, 0.61, 0.69, 0.74, 0.77, 0.79, 0.8, 0.8, 0.8, 0.8 $\sigma = 0.1$ |
| $\delta_l$ | Larval transformation probability | Uniform (0, 1) | Beta ($\alpha, \beta$) $\mu_{0-3} = 0.05$, $\mu_4 = 0.515$, $\mu_{5-10} = 0.9$ $\sigma = 0.1$ |
| $\mu_{entry.ocean}$ | Ocean entry survival | Uniform (0, 1) | Beta ($\alpha, \beta$), $\mu = 0.46$, $\sigma = 0.1$ |
| $\zeta_j$ | Juvenile swim upriver probability (by ocean year; e.g., SUR Juv-6 in Figs 3 and 4 is equivalent to $\zeta_6$ in math notation) | Uniform (0, 1) | Beta ($\alpha, \beta$) $\mu_{0-2} = 0.05$, $\mu_3 = 0.1$, $\mu_4 = 0.15$, $\mu_5 = 0.45$, $\mu_{6-10} = 0.9$ $\sigma = 0.1$ |
| $\mu_{yearly.ocean}$ | Yearly ocean survival | Uniform (0, 1) | Beta ($\alpha, \beta$), $\mu = 0.7$, $\sigma = 0.1$ |
| $\mu_{entry.river}$ | River entry (river mouth) survival | Uniform (0, 1) | Beta ($\alpha, \beta$), $\mu = 0.67$, $\sigma = 0.1$ |
| $\eta_{0-1}$ | Spawn probability (years 0 and 1) | Uniform (0, 1) | Beta ($\alpha, \beta$), $\mu_0 = 0.05$, $\mu_1 = 0.7$, $\sigma = 0.1$ |

**Table 3. Case study management scenario examples.**

| Scenario | Life cycle parameters | Barrier (probability of passage) | # Adults translocated | # Hatchery transformers |
|---|---|---|---|---|
| 1 | Default | 0 | 0 | 0 |
| 2 | Default | 1 (0.6) | 0 | 0 |
| 3 | Default | 1 (0.6) | 100 | 0 |
| 4 | Default | 1 (0.6) | 0 | 1000 |
| 5 | Default | 1 (0.6) | 100 | 1000 |

upstream barrier with 60% passability (relative to scenario 2; Fig 6). Scenario 5 includes adult translocations and hatchery releases of transformers simultaneously; the number of spawning lamprey in this scenario appears to roughly match the baseline conditions without any upstream barriers (compare to scenario 1 in Fig 6).

## Discussion

Our LLCM is the first step in building a tool that could 1) inform conservation efforts of sensitive populations of lamprey species and 2) inform control efforts of invasive lamprey populations of in the Laurentian Great Lakes region. The current best use for the LLCM is as a heuristic tool and a means to generate hypotheses to further explore. For demonstration purposes, we focused on Pacific lamprey as a case study using expert knowledge and literature to inform the model. We provide custom R functions, a Shiny application (https://rconnect.usgs.gov/LampreyLCM/), and all underlying code to allow future re-use and full adaptation of the model [23]. We demonstrate model stability with current default conditions

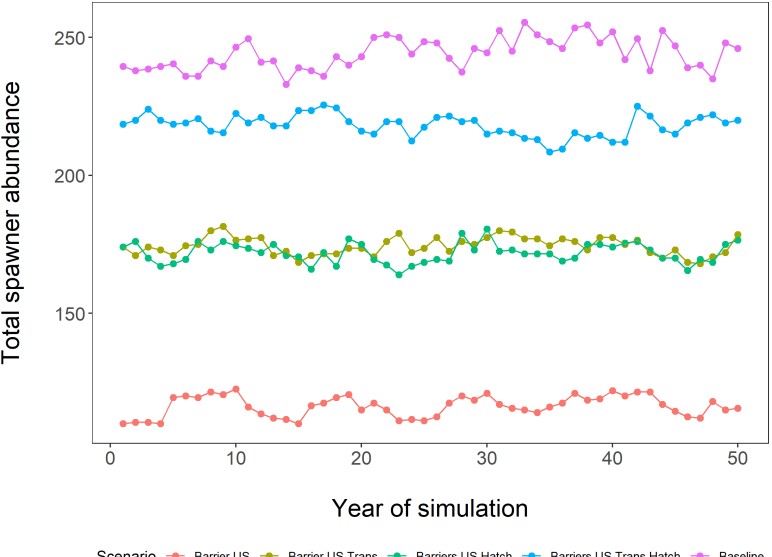

**Fig 6. Hypothetical management scenarios as examples of model utility.** Scenario 1 (pink): "Baseline" (default parameters, including no barriers). Scenario 2 (red): "Barrier.US" addition of upstream barrier with 60% passability. Scenario 3 (olive): "Barrier.US.Trans" same as Scenario 2 + translocating 100 adults (when available) above the barriers. Scenario 4 (green): "Barriers.US.Hatch" same as Scenario 2 + 1000 hatchery transformers released into river. Scenario 5 (blue): "Barriers.US.Trans.Hatch" combination of Scenario 3 and 4; that is, upstream barrier with 60% passability + translocating 100 adults (if available) above the barriers + 1000 hatchery transformers released into river. Here, we are focusing on the relative values between scenarios to explore management alternatives.

and provide examples of functionality through both a global sensitivity analysis and management scenario simulations. Below we discuss model limitations and interpretations of the global sensitivity analyses and the management scenario simulations.

## Model limitations

Process-based models are useful tools in exploring connected systems, complex ecological phenomena, and whole-ecosystem management [42,71,72]. Yet, all models are necessarily simplified approximations of reality. Structurally, there are a few model limitations. The current model assumes a closed population. That is, lampreys do not emigrate from nor immigrate to other populations. If lamprey do not pass a barrier in the model, they do not contribute to the closed population. Similarly, adult translocations in the model do not consider that in the real world, depending on the collection and release points, lamprey may be diverted from one tributary or population to another. Thus, the LLCM is currently limited in not being able to explore meta-population dynamics with different scenarios in adjacent, connected streams. While the addition of a meta-population module can be added to future versions of the LLCM, this simplifying assumption may or may not be an important omission, depending on the research questions, the ecology and population dynamics of the particular lamprey species or population in question. If watersheds are not net population sources nor sinks at the scale of interest, the modeled population dynamics likely will yield similar results, despite ignoring emigration and immigration. This limitation is also likely to vary spatially, considering larger areas will likely lead to neutral net emigration and immigration into and out of the model domain (e.g., modeling the global population versus local populations).

For the sake of simplicity and computational efficiency, we ignored growth and biomass in the LLCM, and instead focused on tracking individual lamprey numbers. This has several important consequences worth considering. Fecundity

is a function of lamprey size [73,74], yet since size is not considered in the model, all individual fecundity values are drawn from the same distribution (regardless of pre-spawn instream holding time, which may or may not affect fecundity in reality). LLCM users can currently explore the effects of adult body size on spawning populations by altering the fecundity values directly to represent scenarios of differing body size. Future versions of the LLCM could incorporate more complexity in this process. For example, body size is likely determined by growth during larval and juvenile stages, which might be driven by water temperatures [75–78], food availability [49,74,79–81], and density [50,82–85]. The underlying modeling framework of the LLCM is flexible enough to handle additional processes, provided that these can be described mathematically.

Larval body size also has important implications for density-dependent growth and survival. As larvae grow each year, they will use more of the limited available habitat [14,86]. Currently, all individuals of age-1+larvae are treated equally in the calculation of carrying capacity (and thus density-dependent survival). This simplifying assumption could be improved in future model versions to estimate the unequal contributions of each size or age-class to density-dependent survival.

We were able to find default parameters from Pacific lamprey-specific work for many, but not all parameters. Life cycle parameters in Table 1 came from different populations and, sometimes, different species. While these default values are a useful starting place, some parameter values will likely vary for different populations and species of lamprey. Future LLCM users should attempt to parameterize important differences for populations of interest. Notably for Pacific lamprey, we were unable to find larval freshwater survival and transformation probability values, juvenile survival values upon entering and exiting the ocean, and yearly ocean survival values. This is not surprising given the difficulty of marking or tracking wild lamprey in either the sediment or in the open ocean. Interestingly, these parameters appeared to be important in the global sensitivity analyses (see below).

## Global sensitivity analysis

We employed two strategies for exploring parameter values and variable influence. In the first GSA we allowed a broad range of possibilities for parameter values. Some parameter values are highly uncertain, so exploring the entire range of possibilities can be a more conservative approach. However, this means that we explored potentially unrealistic values (e.g., survival parameters close to 0 or 1). Thus, in the second GSA, constraining values to a more realistic range will shift the 'importance' (see Methods) weighting to other variables since the influence of some variables will become more restricted (e.g., no longer uniform distribution from 0 to 1). Thus, we are more inclined to trust the results of the second GSA, while recognizing that the assigned uncertainty distributions we used to explore parameter space might not fully capture the realm of possibilities across all populations or species of lamprey. The range of explored parameter values for each variable affected the results (Figs 4 and 5) as different ranges can highlight different sensitivities and, thus, importance metrics. However, it is not always clear why particular variables become more or less important due to complications of variable interactions in a GSA. That is, when variables interact with each other, the importance of a single variable can change depending on the parameter values of other variables. This analysis highlights the usefulness of continued monitoring to better estimate the life cycle parameter values and variability for populations of interest, since these inputs can change the relative importance of each parameter in the life cycle model. Our hope is that future model users continue to explore these parameters and their influence on population dynamics.

In the GSA with the broadest explored parameter space, survival values for juveniles entering the ocean and the river, and yearly ocean survival values were all influential, whereas ocean entry survival was not influential in the more restricted parameter space analysis. All three of these parameters are not well-resolved in the empirical lamprey literature, such that we used Chinook salmon values for two of them and based the third on assumptions. While these are difficult values to estimate, the GSA suggests that they would be useful in being able to predict spawning lamprey numbers, especially yearly ocean survival, which came out in the top 5 most influential parameters in both GSAs.

Early survival values were highly influential in each GSA. Age-0 larval survival was the most influential parameter in the broad GSA (Fig 4), while egg survival was the most influential in the restricted parameter GSA (Fig 5). Both age-0 larval survival and egg survival default values are based here on some empirical literature and assumptions. Other larval survival (and transformation probability) parameters were also influential in both analyses, although the two GSAs differed in which larval age-classes had the most influential effects on predicting spawners. In the broad GSA, younger age-classes of larvae had a large range of survival values (i.e., between zero and one), which can lead to large differences in simulated population sizes since these initial numbers (e.g., age-0 and age-1 larval survival in Fig 4) impact subsequent age classes. In the more restricted GSA (Fig 5), survival values for age-0 and age-1 larvae were less important because they were restricted to more reasonable, lower values with less variation. Larval survival and transformation probabilities were adjusted to match an age-class distribution of transformers from a single stream, but otherwise were not constrained by real data. More information on all of the above parameters would help to resolve a Pacific lamprey-specific life cycle model.

Fecundity was one of the few parameters that was consistent across GSAs (Table 2), as it is relatively well-known and, thus, constrained. This parameter was influential in both GSAs, however, the magnitude of importance changed substantially between analyses (Figs 4 and 5). These analyses suggest that better constraining all parameters can alter, and potentially help elucidate, which parameters are influential. These simulations highlight the value of fine-tuning life cycle model parameters using field and laboratory research. Future uses of the LLCM can include more systematically altering the parameter mean values and variation to understand the effect that fine-tuning any particular parameter has on parameter influence in the model.

## Management scenario sensitivity analysis

Our management scenarios display a few simple examples of what the LLCM can explore. We focused on two management strategies that have been highlighted as potentially useful for the conservation and restoration of Pacific lamprey [30,66,87]. In Fig 6, the population of spawners stabilizes at various values given the management scenarios employed and the default values used for all other parameters. The absolute number of lamprey spawners may not be meaningful outside the context of the particular values selected for model life cycle parameters. For example, if arbitrary parameters (such as larval survival or carrying capacity values) are selected that do not represent any particular stream, the absolute numbers in the model output may not necessarily be biologically meaningful. Yet, even in these cases where absolute numbers are not calibrated with empirical data, relative differences amongst management scenarios can be informative as we compare the effects of modifying different processes (which makes the assumption that we have mathematically represented the processes accurately).

In these case studies, we demonstrated that the barrier addition, fish translocation, and hatchery release operations are functioning as designed in the LLCM. The barrier addition reduced the spawning population relative to baseline conditions, which was expected. Each of the adult translocation and the hatchery release of transformers options appeared to increase the spawning population to some degree, although either strategy in isolation was not enough to overcome the deficit from the addition of one upstream barrier. These two strategies in this simplistic simulation appeared to be equal in efficacy, although note an order of magnitude difference in the number of transformers released from hatcheries (1000) relative to the maximum number of translocated adults (100). Our last scenario demonstrates the additive effects of including both adult translocations and hatchery releases of transformers simultaneously, in which case the deficit from the addition of an upstream barrier appears to be offset. We caution against generalizing these results and basing management decisions upon these simple scenarios. The default parameters we have used throughout the life cycle model are only an approximation of any one Pacific lamprey population, and the broad uncertainty in the results needs to be reduced through future study.

## Conclusions and future directions

Management actions, climate, and life cycle parameters can interact in unexpected ways, which can be explored with our life cycle model. Management actions can be financially costly, which typically results in only a limited number of methods being available to any one conservation group or effort. Understanding the expected efficacy of such alternative actions, before taking such actions, can be greatly beneficial to species conservation. Future studies using this model could explore tradeoffs between barriers (including multiple barriers in both the upstream and downstream directions), fish translocation, and hatchery releases amidst a changing climate. Users of the model can adjust summer and winter mortality and carrying capacities to explore what future climate scenarios might bring to lamprey populations. In addition to conservation efforts, this model could prove useful in assisting the control of the invasive sea lampreys, such as in the Laurentian Great Lakes [36]. We hope that this modeling framework is useful in aiding hypothesis generation and directing future research efforts, with the ultimate goal of improving lamprey conservation and management.

## Acknowledgments

We thank Burke Strobel, Mariah Mayfield, Hunter J. Cole, Tom Stahl and the USGS Viz Lab for helpful feedback on the design and layout of the Shiny application. This manuscript benefitted from a review by Bob Rose, Megan Sabal, and Nicholas Schloesser. Monica R. Blanchard provided the beautiful illustration used in Fig 1. Thomas Hobbs and Mevin Hooten provided a Bayesian Modeling course to DGEG, which was helpful to understand and document probability density/mass functions, moment matching, and math notation. Any use of trade, firm, or product names is for descriptive purposes only and does not imply endorsement by the US government.

## Author contributions

**Conceptualization:** Dylan G. E. Gomes, Joseph R. Benjamin, Benjamin J. Clemens, Ralph Lampman, Jason B. Dunham.

**Data curation:** Dylan G. E. Gomes, Benjamin J. Clemens, Ralph Lampman.

**Formal analysis:** Dylan G. E. Gomes.

**Funding acquisition:** Joseph R. Benjamin, Benjamin J. Clemens, Jason B. Dunham.

**Investigation:** Dylan G. E. Gomes.

**Methodology:** Dylan G. E. Gomes.

**Project administration:** Joseph R. Benjamin.

**Resources:** Benjamin J. Clemens.

**Software:** Dylan G. E. Gomes.

**Supervision:** Joseph R. Benjamin.

**Visualization:** Dylan G. E. Gomes, Ralph Lampman.

**Writing – original draft:** Dylan G. E. Gomes.

**Writing – review & editing:** Dylan G. E. Gomes, Joseph R. Benjamin, Benjamin J. Clemens, Ralph Lampman, Jason B. Dunham.

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
