## [Decision Letter · Decision Letter 0]

28 Jan 2025

PONE-D-24-54766New technology for an ancient fish: A lamprey life cycle modeling tool with an R Shiny applicationPLOS ONE

Dear Dr. Gomes,

Thank you for submitting your manuscript to PLOS ONE. After careful consideration, we feel that it has merit but does not fully meet PLOS ONE’s publication criteria as it currently stands. Therefore, we invite you to submit a revised version of the manuscript that addresses the points raised during the review process.

We look forward to receiving your revised manuscript.

Kind regards,

Hector Escriva, PhD

Academic Editor

PLOS ONE

Journal Requirements:

2. Please note that PLOS ONE has specific guidelines on code sharing for submissions in which author-generated code underpins the findings in the manuscript. In these cases, we expect all author-generated code to be made available without restrictions upon publication of the work. 

Please review our guidelines at https://journals.plos.org/plosone/s/materials-and-software-sharing#loc-sharing-code and ensure that your code is shared in a way that follows best practice and facilitates reproducibility and reuse.

3. Please note that funding information should not appear in the Acknowledgments section or other areas of your manuscript. We will only publish funding information present in the Funding Statement section of the online submission form. Please remove any funding-related text from the manuscript. 

4. Please note that your Data Availability Statement is currently missing the repository name. If your manuscript is accepted for publication, you will be asked to provide these details on a very short timeline. We therefore suggest that you provide this information now, though we will not hold up the peer review process if you are unable.

5. We note that Figure 1 in your submission contain copyrighted images. All PLOS content is published under the Creative Commons Attribution License (CC BY 4.0), which means that the manuscript, images, and Supporting Information files will be freely available online, and any third party is permitted to access, download, copy, distribute, and use these materials in any way, even commercially, with proper attribution. For more information, see our copyright guidelines: http://journals.plos.org/plosone/s/licenses-and-copyright.

1) You may seek permission from the original copyright holder of Figure 1 to publish the content specifically under the CC BY 4.0 license. 

2) If you are unable to obtain permission from the original copyright holder to publish these figures under the CC BY 4.0 license or if the copyright holder’s requirements are incompatible with the CC BY 4.0 license, please either i) remove the figure or ii) supply a replacement figure that complies with the CC BY 4.0 license. Please check copyright information on all replacement figures and update the figure caption with source information. 

If applicable, please specify in the figure caption text when a figure is similar but not identical to the original image and is therefore for illustrative purposes only.

6. Please upload a new copy of Figures 3 and 4 as the detail is not clear. Please follow the link for more information:

https://blogs.plos.org/plos/2019/06/looking-good-tips-for-creating-your-plos-figures-graphics/

https://blogs.plos.org/plos/2019/06/looking-good-tips-for-creating-your-plos-figures-graphics/

7. Please include a copy of Tables 4 and 5 which you refer to in your text on pages 19-20.

**Additional Editor Comments:**

We would particularly appreciate it if you could respond thoroughly to the comments of Reviewer 1 and revise the text according to the considerations of both reviewers.

Reviewers' comments:

Reviewer's Responses to Questions

**Comments to the Author**

1. Is the manuscript technically sound, and do the data support the conclusions?

Reviewer #1: Yes

Reviewer #2: Yes

2. Has the statistical analysis been performed appropriately and rigorously? 

Reviewer #1: Yes

Reviewer #2: Yes

3. Have the authors made all data underlying the findings in their manuscript fully available?

Reviewer #1: Yes

Reviewer #2: Yes

4. Is the manuscript presented in an intelligible fashion and written in standard English?

Reviewer #1: Yes

Reviewer #2: Yes

5. Review Comments to the Author

Reviewer #1: This manuscript submitted by Dylan G. E. Gomes et al. presents a life cycle model for Lampreys (Petromyzontiformes) development using R shiny. The model, named as lamprey life cycle model (LLCM), was based on a simulation-based framework over 100 years. The default parameter values of lamprey life cycles were derived from references contains empirical data or based on assumptions. Then, the authors conducted a global sensitivity analysis (GSA) and Random Forest analysis to evaluate the influence of individual life stage parameters on lamprey population size and identify critical knowledge gaps. Additionally, they analyzed five management scenarios to illustrate potential management applications, including the addition of an upstream barrier, adult translocations, and hatchery releases. Overall, this manuscript developed the lamprey life cycle model to support conservation and control efforts for lamprey species. It is a heuristic tool for generating hypotheses and includes custom R functions, a Shiny application, and underlying code for future use and adaptation. However, there is a need to tone down some claims that are not fully supported by the data presented.

(1) Environmental variables such as temperature and density can influence fish sex determination. Study in 2017 (https://doi.org/10.1098/rspb.2017.0262) suggested that density and nutrition can affect the sex ratio in sea lampreys. Therefore, the default value of sex ratio parameters maybe adjusted, environmental factors need to be considered. Additionally, whether user-defined parameters and default values can be adjusted based on observed field data from other lamprey species, especially those were not defined in Pacific lamprey.

(2) The author suggests that the parameters of LLCM can be adjusted with field data, ecological theory, expert opinion, and sensitivity analysis, which indicated that the model was highly adaptable in the face of uncertainty and had potential value in the conservation and management of lamprey species. Although the simulation results provide useful guidance for selecting management strategies, how can the model to be adjusted using new empirical data to improve its application and decision-support capabilities in the future.

(3) What is the reason for the differed results of GSA and GSA forest analyses? The results were differences between broad and realistic parameter sampling schemes, but the reasons behind the shifts in parameter sensitivity (such as why age-7 larval survival becomes more important?) are not discussed. Briefly explain or hypothesize why certain parameters gain or lose influence between two analyses.

(4) In management cenario sensitivity results, why do they select five scenarios? Although the author presented that barriers reduce spawning populations and translocations restore them, the implications for management aren't deeply discussed. For instance, how feasible are the interventions? A brief discussion of the practical challenges or advantages of implementing scenarios like translocations and hatchery supplementation.

(5) Can LLCM be applied to the life cycle analysis of other lamprey species, such as the Japanese lamprey or other marine species?

(6) Is there any research data on the population abundance of Pacific lamprey, and has this data been incorporated into the model?

(7) The reference format is inconsistent, such as in lines 866-867 and lines 874-876.

Reviewer #2: Overall comments

A well written paper describing a useful modeling tool. Two general comments and a few line-by-line edits

The GSA analysis – a plain language description of why this analysis was done would prepare the reader for considering the implications in the discussion. The MSE response variable throws me off as the response of interest for a sensitivity analysis. More detail on what this variable is comparing would be helpful. With a sensitivity analysis, I think of a measure of how much the response variable of interest changes (in this case the size of the spawner population) as we vary a parameter. I have a hard time placing your analysis in this conceptual framework. Not to say that it needs to be, but currently this section is hard for me to follow (both the methods and the rationale for why it is important).

Applications to Great Lakes sea lamprey control – this is mentioned in the first sentence of the discussion and in the conclusion, but the idea does not get developed at all in the text. If this is an important part of this work, the idea should be developed more in the text. An example of how it could be used or description of the management need that could be addressed would be helpful here. Similarities and differences to other DSTs that have been used to model sea lamprey dynamics in the Great Lakes would also be relevant to this discussion. As a starting point, Jones et al. (2009) and subsequent applications of this model would be relevant to discuss.

Jones, M.L., Irwin, B.J., Hansen, G.J.A., Dawson, H.A., Treble, A.J., Liu, W., Dai, W., Bence, J.R., 2009. An Operating Model for the Integrated Pest Management of Great Lakes Sea Lampreys. TOFISHSJ 2, 59–73. https://doi.org/10.2174/1874401X00902010059

Line by line comments

Figure 2 Caption.

Between and within year descriptions – from the model description,

Line 330 – burn in of 50 years in Fig. 3, doesn’t match description in text (few years)

Figure 3 Captions- I find the last sentence of the caption confusing/don’t follow the rationale. In cases where carrying capacity is based on data or even expert opinion for a particular stream, seems like the number of spawners is relevant model output.

Lines 494 – mean squared error response – what is being compared to what here? I usually think of using MSE to compare an estimated value to a known value. I get lost in the description of how it is applied here, I am not familiar with using MSE as a metric to describe a sensitivity analysis. Seems like a crucial concept to understand the significance of the whole GSA section and it gets lost on me. Some added plain language text on the rationale for the methods and why the analysis was done would be helpful.

Lines 612 – modeling local populations also become important when modeling the combined outcome form multiple stream-specific conditions/management actions. This allows for more realistic simulations.

Lines 697 –Is the point here that there will tend to be considerable uncertainty in life history parameters, particularly carrying capacity terms, so one shouldn’t put much weight on the absolute numbers of spawners as this is an equilibrium-type model (gets spun up with a burn-in period, rather than initialized at a specific value)? This could be stated more plainly. Also, if this is the case, why should we consider comparisons among management scenarios (relative number of spawners) as reliable but absolute numbers as unreliable? Digging into this question a bit more would help the reader here.

Lines 730 – an example of how it could help control efforts in the Great Lakes would be useful. This is brought up in the first line of the discussion as well, but never gets fleshed out with any detail.

6. PLOS authors have the option to publish the peer review history of their article (what does this mean? ). If published, this will include your full peer review and any attached files.

**Do you want your identity to be public for this peer review?** For information about this choice, including consent withdrawal, please see our Privacy Policy .

Reviewer #1: No

Reviewer #2: No

---

## [Author Response · Author response to Decision Letter 1]

27 Feb 2025

Reviewer #1: This manuscript submitted by Dylan G. E. Gomes et al. presents a life cycle model for Lampreys (Petromyzontiformes) development using R shiny. The model, named as lamprey life cycle model (LLCM), was based on a simulation-based framework over 100 years. The default parameter values of lamprey life cycles were derived from references contains empirical data or based on assumptions. Then, the authors conducted a global sensitivity analysis (GSA) and Random Forest analysis to evaluate the influence of individual life stage parameters on lamprey population size and identify critical knowledge gaps. Additionally, they analyzed five management scenarios to illustrate potential management applications, including the addition of an upstream barrier, adult translocations, and hatchery releases. Overall, this manuscript developed the lamprey life cycle model to support conservation and control efforts for lamprey species. It is a heuristic tool for generating hypotheses and includes custom R functions, a Shiny application, and underlying code for future use and adaptation. However, there is a need to tone down some claims that are not fully supported by the data presented.

(1) Environmental variables such as temperature and density can influence fish sex determination. Study in 2017 (https://doi.org/10.1098/rspb.2017.0262) suggested that density and nutrition can affect the sex ratio in sea lampreys. Therefore, the default value of sex ratio parameters maybe adjusted, environmental factors need to be considered. Additionally, whether user-defined parameters and default values can be adjusted based on observed field data from other lamprey species, especially those were not defined in Pacific lamprey.

Thank you for this point. We have added this reference and some language building on these points to the section on “Spawning and fecundity” within the “Default parameter values” subsection, as potential deviations from the default parameter value of 0.5. Users are free to set any parameter to whatever value they choose. We have further suggested that users can select their own values based on observed field data, including from other lamprey species in the last sentence of the introduction. “However, lamprey managers and researchers can adjust parameter values based on empirical data and expert opinion to represent a life cycle model for any species of lamprey.” We also state in the methods “For modeling other lamprey species of interest (e.g., sea lamprey, Petromyzon marinus), the “ocean” labels can simply represent bodies of fresh water, such as the Great Lakes. The user needs only to replace the relevant life cycle parameters in the model to represent different species of lamprey.”

(2) The author suggests that the parameters of LLCM can be adjusted with field data, ecological theory, expert opinion, and sensitivity analysis, which indicated that the model was highly adaptable in the face of uncertainty and had potential value in the conservation and management of lamprey species. Although the simulation results provide useful guidance for selecting management strategies, how can the model to be adjusted using new empirical data to improve its application and decision-support capabilities in the future.

The model parameters can be modified at any point in time to reflect new empirical data or alternative management strategies (see points above for text in the manuscript reflecting this point). Additionally, the code is openly available for anyone who wishes to build out additional processes that arise from new knowledge of lamprey population dynamics or interactions with other species and environmental conditions.

(3) What is the reason for the differed results of GSA and GSA forest analyses? The results were differences between broad and realistic parameter sampling schemes, but the reasons behind the shifts in parameter sensitivity (such as why age-7 larval survival becomes more important?) are not discussed. Briefly explain or hypothesize why certain parameters gain or lose influence between two analyses.

The range of values chosen for each parameter can affect the results. Different ranges can highlight different sensitivities and importance metrics. Yet it isn’t always clear why particular variables become more or less important because it is also complicated by parameter interactions in a global sensitivity analysis. That is, when parameters interact with each other, the importance of a single parameter can change depending on the values of other parameters. This can lead to complicated patterns in differences in the importance metrics. We’ve added this language to the first paragraph of the Discussion subsection labelled “Global sensitivity analysis.” We additionally add language around potential explanations for differences in larval survival importance within the same subsection. The changed paragraph (see other paragraphs for more information) now reads (additions in bold):

“Early survival values were highly influential in each GSA. Age-0 larval survival was the most influential parameter in the broad GSA (Figure 4), while egg survival was the most influential in the restricted parameter GSA (Figure 5). Both age-0 larval survival and egg survival default values are based here on some empirical literature and assumptions. Other larval survival (and transformation probability) parameters were also influential in both analyses, although the two GSAs differed in which larval age-classes had the most influential effects on predicting spawners. In the broad GSA, younger age-classes of larvae were allowed to have survival values between zero and one. These values can lead to large differences in potential population sizes since these initial numbers (e.g., age-0 and age-1 larval survival in Figure 4) impact all other subsequent age classes. In the more restricted GSA (Figure 5), age-0 and age-1 larval survival values were perhaps less important because they were restricted to more reasonable, lower values with less variation. Larval survival and transformation probabilities were adjusted to match an age-class distribution of transformers from a single stream, but otherwise were not constrained by real data. More information on all of the above parameters would help to resolve a Pacific lamprey-specific life cycle model.”

(4) In management cenario sensitivity results, why do they select five scenarios? Although the author presented that barriers reduce spawning populations and translocations restore them, the implications for management aren't deeply discussed. For instance, how feasible are the interventions? A brief discussion of the practical challenges or advantages of implementing scenarios like translocations and hatchery supplementation.

We have added language in the “Management scenario case study examples” subsection within the methods. This addition reads: “We selected these scenarios to broadly represent common management actions relating to lamprey conservation and/or control [60]. These include the influence of barriers [61–63], translocations [30,64], and hatchery releases [65,66] on lamprey populations.” These scenarios are intended to be examples of what the model is capable of. They are case studies and aren’t intended to be used to imply management decisions in this context. We are not advocating for or against any particular management intervention, as this is outside the scope of this manuscript.

(5) Can LLCM be applied to the life cycle analysis of other lamprey species, such as the Japanese lamprey or other marine species?

Absolutely. We have added a sentence at the end of the introduction, which now reads:

“Our model and tool [23] has the potential for continued modification to fit the various uses of the lamprey management community. As a specific case-study to demonstrate potential uses and benefits of such a modeling framework, we focus on using published information and best professional knowledge to inform parameter estimates for the Pacific lamprey, Entosphenus tridentatus. However, lamprey managers and researchers can adjust parameter values based on empirical data and expert opinion to represent a life cycle model for any species of lamprey.”

This is also mentioned in the manuscript in the Default parameter values as:

“…it is up to the user to specify the parameters that most closely represents their system and lamprey species of interest. Sometimes information is limited, and one will have to make assumptions about how to use information from other populations or species. Models are dependent on information to describe interactions, relationships, and other processes, and process-based models, in particular, can incorporate a broad range of information types [39]. While empirical data are often used to parameterize key model processes, these mathematical models are flexible such that gaps in knowledge can be represented by ecological theory, expert opinion, educated guesses, or informed by sensitivity analyses (see the Global sensitivity analysis section below) that explore the entire range of realistic possibilities. We provide a description of default parameters and where they come from (Table 1) but realize that these parameters may not be appropriate for every population of Pacific lamprey or other lamprey species.”

And in the last paragraph of the conclusion as:

“In addition to conservation efforts, this model could prove useful in assisting the control of the invasive sea lampreys, such as in the Laurentian Great Lakes [36]. We hope that this modeling framework is useful in aiding hypothesis generation and directing future research efforts, with the ultimate goal of improving lamprey conservation and management.”

(6) Is there any research data on the population abundance of Pacific lamprey, and has this data been incorporated into the model?

Thank you for this question. Population abundances of Pacific lamprey will vary from stream to stream and depends on the scale of interest (e.g., reach, entire stream, entire basin, etc.). The model is a simulation-based heuristic or management strategy evaluation model, rather than a statistical model that can be fit to empirical data. Thus, we are not convinced that providing a number of adults, for example, based on one particular stream reach will be particularly useful to users of the model. With this said, the users can adjust unknown parameter values to try and calibrate the model to their population of interest, but there is no way for us to do that for them. We also believe that users should not focus too much on absolute abundance (see other responses). What is more interesting for evaluating alternative management actions is the relative differences among scenarios. If you bypass 1000 adults, are populations higher than if you just increase fish passage by 10%? The absolute numbers are not likely going to be accurate, but the relative differences might be informative.

(7) The reference format is inconsistent, such as in lines 866-867 and lines 874-876.

Lines 866-867 span two references, the first of which is a peer-reviewed article and the second of which is a whitepaper. We do not see inconsistencies with these two with respect to journal guidelines. We will refer to typesetters upon publication to assist us with clarifying this. Lines 874-876 also spans two references, the second of which was missing the journal information. We have added this information to this reference. Thank you for catching this.

Reviewer #2: Overall comments

A well written paper describing a useful modeling tool. Two general comments and a few line-by-line edits

Thank you for your positive feedback

The GSA analysis – a plain language description of why this analysis was done would prepare the reader for considering the implications in the discussion. The MSE response variable throws me off as the response of interest for a sensitivity analysis. More detail on what this variable is comparing would be helpful. With a sensitivity analysis, I think of a measure of how much the response variable of interest changes (in this case the size of the spawner population) as we vary a parameter. I have a hard time placing your analysis in this conceptual framework. Not to say that it needs to be, but currently this section is hard for me to follow (both the methods and the rationale for why it is important).

Thank you for bringing this up. We have added the following language to the section that introduces the global sensitivity analyses:

“Sensitivity analyses allow for an exploration of model sensitivities to particular parameters. As opposed to altering one parameter at a time, and keeping all others at fixed values, global sensitivity analyses (GSA) allow an assessment of parameter influence on overall model dynamics while considering all possible interactions among parameters [58,59]. A GSA is used to evaluate how variations in input parameters influence the uncertainty in the output of the model, which helps identify the inputs that have the most significant impact, assess interactions among variables, and determine whether certain inputs jointly affect the results. By understanding these sensitivities, GSA improves model robustness, ensuring it is not overly sensitive to small changes that could indicate instability. It also guides data collection efforts by prioritizing key parameters that significantly affect predictions. Additionally, GSA supports decision-making in areas like risk assessment, optimization of management strategies, and policy-making by clarifying how different inputs drive outcomes. Furthermore, GSA aids in validating model assumptions by verifying whether simplifications or approximations are justified based on the sensitivity of inputs. Unlike local sensitivity analysis, which examines small perturbations around a fixed point, GSA evaluates input variations across their entire range, making GSA a more comprehensive approach for analyzing complex systems. A GSA is advantageous to fixing all non-modified parameter values to mean values and assuming that they are correct, because parameter space, and thus parameter influence, is more thoroughly explored.”

Regarding “With a sensitivity analysis, I think of a measure of how much the response variable of interest changes (in this case the size of the spawner population) as we vary a parameter.” – this is exactly what we are measuring here with this importance metric. For regressions, this is commonly computed as MSE on the data that are left out of training. Then a % change in this MSE is computed after permuting each variable randomly. The question we are addressing here is if you randomize a predictor variable, how much worse is the model at predicting out of sample data? If it gets a lot worse (higher % increase in MSE), that suggests that it is a useful variable (high importance). If % increase in MSE for one variable is more than any other variable this suggests that this is the most useful variable for understanding the response of interest (spawners, in this case). We’ve added to the language in the methods, which now reads:

“We then used the function `importance` to determine the importance of each of the 44 variables, as measured by the percent increase in mean squared error (MSE) when each variable is included. The MSE on the out-of-bag (OOB) portion of the data (data not included in training the model) is calculated for each tree, then the same is done after permuting each variable in the model. The difference in MSE values between the initial regression forest and the permutation of each variable are averaged across all trees and normalized by the standard deviation of the differences to generate a percent change in MSE when a given variable was permuted. A higher percent increase in MSE suggests that a randomly permuted variable is more important/influential than another randomly permuted variable that leads to lower changes is MSE.”

We hope that clarifies the GSA.

Applications to Great Lakes sea lamprey control – this is mentioned in the first sentence of the discussion and in the conclusion, but the idea does not get developed at all in the text. If this is an important part of this work, the idea should be developed more in the text. An example of how it could be used or de

---

## [Editor Report · Decision Letter 1]

8 Apr 2025

New technology for an ancient fish: A lamprey life cycle modeling tool with an R Shiny application

PONE-D-24-54766R1

Dear Dr. Gomes,

We’re pleased to inform you that your manuscript has been judged scientifically suitable for publication and will be formally accepted for publication once it meets all outstanding technical requirements.

Kind regards,

Hector Escriva, PhD

Academic Editor

PLOS ONE
---

## [Editor Report · Acceptance letter]

PONE-D-24-54766R1

PLOS ONE

Dear Dr. Gomes,

I'm pleased to inform you that your manuscript has been deemed suitable for publication in PLOS ONE. Congratulations! Your manuscript is now being handed over to our production team.

Kind regards,

on behalf of

Dr. Hector Escriva

Academic Editor

PLOS ONE